# Dynamic Expression of Long Non-Coding RNAs Throughout Parasite Sexual and Neural Maturation in *Schistosoma Japonicum*

**DOI:** 10.3390/ncrna6020015

**Published:** 2020-04-01

**Authors:** Lucas F. Maciel, David A. Morales-Vicente, Sergio Verjovski-Almeida

**Affiliations:** 1Laboratório de Expressão Gênica em Eucariotos, Instituto Butantan, São Paulo SP 05503-900, Brazildavid.vicente@butantan.gov.br (D.A.M.-V.); 2Programa Interunidades em Bioinformática, Instituto de Matemática e Estatística, Universidade de São Paulo, São Paulo SP 05508-900, Brazil; 3Departamento de Bioquímica, Instituto de Química, Universidade de São Paulo, São Paulo SP 05508-900, Brazil

**Keywords:** parasitology, lncRNAs, WGCNA, gene co-expression network, synteny, nervous system

## Abstract

*Schistosoma japonicum* is a flatworm that causes schistosomiasis, a neglected tropical disease. *S. japonicum* RNA-Seq analyses has been previously reported in the literature on females and males obtained during sexual maturation from 14 to 28 days post-infection in mouse, resulting in the identification of protein-coding genes and pathways, whose expression levels were related to sexual development. However, this work did not include an analysis of long non-coding RNAs (lncRNAs). Here, we applied a pipeline to identify and annotate lncRNAs in 66 *S. japonicum* RNA-Seq publicly available libraries, from different life-cycle stages. We also performed co-expression analyses to find stage-specific lncRNAs possibly related to sexual maturation. We identified 12,291 *S. japonicum* expressed lncRNAs. Sequence similarity search and synteny conservation indicated that some 14% of *S. japonicum* intergenic lncRNAs have synteny conservation with *S. mansoni* intergenic lncRNAs. Co-expression analyses showed that lncRNAs and protein-coding genes in *S. japonicum* males and females have a dynamic co-expression throughout sexual maturation, showing differential expression between the sexes; the protein-coding genes were related to the nervous system development, lipid and drug metabolism, and overall parasite survival. Co-expression pattern suggests that lncRNAs possibly regulate these processes or are regulated by the same activation program as that of protein-coding genes.

## 1. Introduction

Schistosomiasis is a disease caused by parasitic trematodes of the genus *Schistosoma*, which the WHO classifies as a neglected tropical disease [1]. Conservative estimates indicate that at least 230 million people worldwide are infected with *Schistosoma* spp. [2]. *Schistosoma japonicum* is one of the three main species that affects humans. It is prevalent in Asia, primarily China, Indonesia, and the Philippines [2].

This parasite has a very complex life-cycle comprised of several developmental stages, with a freshwater, snail intermediate-host and a final mammalian host [3]. Once the parasites are inside the final host they migrate through blood circulation until they reach the mesenteric veins of the liver, where males and females pair through the gynecophoral canal, in order to promote sexual development and female gonad maturation [4,5]. The paired couples then migrate to the mesenteric veins of the gut, where each female of *S. japonicum* produces 1000–3000 eggs per day [6]. These eggs are released into the bloodstream, where they can actively pass through the intestinal wall and be excreted in the feces, or be carried by the circulation and be trapped in organs, where they cause immunopathologies [7].

Due to the clear importance of sexual maturation for both parasite reproduction and host immunopathogenesis, Wang et al. [8] sequenced the transcriptomes of female and male *S. japonicum* parasites at eight time-points obtained from 14 days post-infection (dpi), before the pairing process, until 28 dpi, when females and males were paired and sexually mature. In this study, the authors obtained valuable information about the reproductive biology of schistosomes, and were able to identify an insect-like hormonal regulatory pathway, along the process of sexual maturation of parasites [8].

Wang et al. [8] also identified novel transcripts that had no annotation because they did not match the known protein-coding genes in their reference dataset, and at the time the authors had not considered the possibility that some of them could actually be long non-coding RNAs (lncRNAs). LncRNAs are defined as transcripts longer than 200 nucleotides, without any apparent protein-coding potential [9]. They were more recently annotated for the first time in *S. japonicum* by Liao et al. [10], using only two RNA-Seq libraries, one from males and one from females. In mammals, lncRNAs were already identified as important regulators of vital processes, including sexual maturation and reproduction [11]. In *S. mansoni,* our group recently showed that some lncRNAs have a tissue-specific expression in ovaries and testis [12], indicating a potential role of lncRNAs in schistosome reproduction. Thus, the RNA-Seq dataset created by Wang et al. [8] represents a valuable, and so far, an unexplored source to identify lncRNAs with dynamic expression throughout sexual maturation in *S. japonicum* and to reveal possible regulatory players in the reproductive biology of these parasites.

The aim of the present work was to annotate a robust and more complete set of lncRNAs in *S. japonicum*, with the pipeline that we developed and previously applied to *S. mansoni* [12]. For this, we used 66 publicly available RNA-Seq libraries from different life-cycle stages, to define a reference set of *S. japonicum* lncRNAs, and we subsequently focused on the 48 libraries from the study of Wang et al. [8] to identify the expression patterns of lncRNAs throughout the sexual maturation process of males and females. Our results provide the basis for future studies on the mechanisms of action of lncRNAs in *S. japonicum* reproductive biology.

## 2. Results

### 2.1. Thousands of lncRNAs are Expressed in S. japonicum

Our pipeline was able to reconstruct 61,298 *S. japonicum* expressed transcripts from the 66 RNA-Seq libraries that were used in this study. The steps to filter pre-mRNAs and mRNAs removed 49,007 transcripts. The remaining 12,291 transcripts, from 7960 different genes, were classified as lncRNAs expressed in *S. japonicum*; on average 1.5 isoforms per lncRNA gene. Of those, 6593 were intergenic lncRNAs (lincRNAs), 4694 were antisense lncRNAs, and 1004 were sense lncRNAs. For comparison, in *S. mansoni*, a total of 633 RNA-Seq libraries were used and 16,000 lncRNAs were identified [12]; the difference in the total number of lncRNAs was probably due to the smaller number of RNA-Seq libraries from different stages and tissues available for transcript reconstruction in *S. japonicum*.

Liao et al. [10] previously identified 3247 and 3033 potential lncRNAs in *S. mansoni* and *S. japonicum*, respectively. These authors used only two RNA-Seq libraries from *S. japonicum*, one from males and one from females [10]. Here we used a much higher number of RNA-Seq libraries from cercariae, sporocysts, schistosomula, and early-development or adult males and females, including the two libraries used by Liao et al. [10], along with the newest version of the genome and transcriptome of *S. japonicum*, which were recently released with significant improvement in assembly contiguity [13]. Our group has recently developed an improved pipeline for identification of lncRNAs [12] and we showed that a large set of the *S. mansoni* transcripts that we previously annotated as lncRNAs with a different pipeline [14] and also that by Liao et al. [10], now seem to represent partially processed pre-mRNAs arising from novel protein-coding genes annotated in the newest version of the *S. mansoni* genome and transcriptome [12]. Considering the limited extent of the *S. japonicum* dataset used by Liao et al. [10], and the difficulties with previous lncRNA identification pipelines [10,14], we decided that in further *S. japonicum* analyses we would only use the 12,291 lncRNAs identified here by our present improved pipeline.

### 2.2. Synteny Conservation is Higher than Gene Sequence Similarity in Schistosoma lncRNA Genes

A blastn search was used to identify sequence similarity between the *S. japonicum* lncRNAs and lncRNAs from other species. Using a relaxed threshold (see Methods), only 1578 *S. japonicum* transcripts (13% of all 12,291 lncRNAs) presented at least one significant hit against *S. mansoni* lncRNAs (Appendix A); on average there were 21.6 *S. mansoni* hits per *S. japonicum* transcript. Among these 1578 *S. japonicum* transcripts had sequence similarity to *S. mansoni* lncRNAs, 503 were lincRNAs (7.6% of all 6593 *S. japonicum* lincRNAs). This was in accordance with the work of Vasconcelos et al. [14], which demonstrated that genomic regions containing lincRNAs in *S. mansoni* showed some sequence conservation among *Schistosoma* species, but showed a lot less conservation than that of the protein-coding genes.

When the same search was performed against human lncRNAs, the number of hits drastically dropped to 19 lncRNAs (Appendix A); on average there were 10.5 human hits per *S. japonicum* transcript. The low number of similar lncRNAs between human and *S. japonicum* was in accordance with the fast evolution identified in lncRNAs, from mammalian species [15].

It has been demonstrated in other species that some lncRNAs show synteny conservation, even when there is a lack of sequence conservation [16]; this might indicate an orthologous relationship between these transcripts and a possible functional conservation [16]. Therefore, we searched for syntenic protein-coding blocks between the *S. japonicum* and *S. mansoni* genomes. In our synteny analysis, we identified 1990 protein-coding genes syntenic blocks covering a representative part of both genomes (Figure 1, green lines). Next, we looked for the lincRNAs that were mapped inside these syntenic blocks and verified that there were 4254 *S. japonicum* lincRNAs, a total of 64% of all lincRNAs identified in *S. japonicum*.

Next, we extracted the orthologous groups of protein-coding genes identified by OrthoMCL (see Methods), and we looked for the closest protein-coding genes upstream and downstream from the 4254 lincRNAs. It was possible to identify 934 distinct *S. japonicum* lincRNAs (14.1% of all 6593 *S. japonicum* lincRNAs), which were not only contained inside the orthologous syntenic blocks but also had at least one equivalent lincRNA in *S. mansoni* with the same pair of closest orthologous protein-coding genes, both upstream and downstream (Appendix A). Given that there were variable numbers of lincRNA isoforms per locus, the number of matching pairs was 3144 (Appendix A), an average of 3.4 matching pairs per 934 distinct *S. japonicum* lincRNAs. The orthologous relationship of these transcripts suggests a possible functional conservation and shows that there were twice as many lincRNAs with syntenic conservation than with sequence similarity between *S. japonicum* and *S. mansoni.*

### 2.3. Gender-Specific lncRNAs Transcriptome

Multidimensional scaling (MDS) analysis (Figure 2a) and the heatmap with all protein-coding and lncRNA transcripts expressed in males and females from 14 to 28 days post-infection (Figure 2b) showed that sexually mature females (22 to 28 days post-infection) had a very different expression profile from immature females (14 to 20 days post-infection) and from males. These results were very similar to the ones presented by Wang et al. [8], even though they had not detected the lncRNAs and they used a different genome and transcriptome version, with a different set of bioinformatic tools.

In fact, when we looked only at the lncRNAs (Figure 3a,b) we could again separate the sexually mature females from the immature females and from males. This suggests that lncRNAs have an activation program similar to that of protein-coding genes, throughout development in *S. japonicum* males and females and that there are sets of lncRNAs that are gender-specific.

It is also similar to the results of Lu et al. [4], which demonstrated in *S. mansoni* that non-paired sexually immature females have a similar expression profile of protein-coding genes to that of males, as the effect of pairing on gene expression was more pronounced in the female worms [4].

### 2.4. Differential Eigengene Network Analysis Reveals Different Patterns of Nervous System Differentiation in Males and Females During Sexual Maturation

The function of the vast majority of lncRNAs in all species is still unknown, but one way to predict function on a genome-wide scale is through the guilt-by-association approach [17,18], with the construction of gene co-expression networks between protein-coding genes and lncRNAs, combined with gene ontology enrichment analyses. In *S. mansoni*, we showed that lncRNAs have a dynamic expression throughout life-cycle progression and are hub genes in the gene co-expression networks [12].

To identify the different *S. japonicum* lncRNAs expression profiles and to assess whether the relationship between consensus modules is preserved in males and females throughout sexual maturation, we performed a differential eigengene network analysis with the weighted gene co-expression network analysis (WGCNA) package [19]. In this approach, gene co-expression networks for males and females were first built separately and then the co-expression modules that were shared by both networks were detected, which were named consensus modules. Consensus modules might represent biological pathways that are shared among the compared conditions, in our case males and females, and a differential relationship between the modules might reveal important differences in pathway regulation [20].

In our set, we were able to identify 11 consensus modules (Figure 4). These modules were composed of dozens to thousands lncRNAs, and the ratio between lncRNAs and mRNAs comprising the modules varied from 24% to 50% (Table 1 and Appendix A).

Each consensus module was then represented by one eigengene, which is the first principal component of the gene expression data of all transcripts comprising that module and is highly correlated with the expression profile of these transcripts [20]. To identify differences in pathway regulation between the sexes, in both male and female co-expression networks we examined the relationship between all consensus module eigengenes, comparing the modules in pairs (Figure 5). Figure 5a,b show the clustering dendrograms of consensus module eigengenes in females and males, respectively, while Figure 5c,f show the correlation between eigengene pairs represented as a heatmap for females and males, respectively. It is possible to see that in the female samples (Figure 5a,c) there were two meta-modules (clusters of highly correlated module eigengenes), one composed by magenta, green, yellow, and blue module eigengenes and the other composed by the remaining module eigengenes. In males (Figure 5b,f), the relationship between the eigengenes and consequently the two meta-modules were not totally preserved. Figure 5e shows the heatmap indicating the preservation of these relationships between the eigengene pairs in females (Figure 5c) and males (Figure 5f). Finally, in Figure 5d we have the preservation measure for each consensus module eigengene; an overall preservation D = 0.62 was obtained, showing that some modules were preserved in both sexes throughout sexual maturation. Overall, the analyses pointed to biological pathways that had a differential regulation in males and females.

Besides the pairwise comparison between the consensus module eigengenes, we found that the eigengenes were correlated to an external trait, which in this case was the number of days post-infection. The correlation with the days is shown in Figure 5 (see “days” label among the modules) but can be better seen in Figure 6, as detailed below.

We found that some module eigengenes had opposite patterns of change as a function of the days post-infection in males and females, such as in the brown module (Figure 6d). In this module, the eigengene value increased in males and decreased in females, as the post-infection days increased (Figure 6d). This was reflected in a positive coefficient of correlation (0.94) with the days of post-infection in males (Figure 6b, brown module) and a negative coefficient of correlation (−0.81) in females (Figure 6a, brown module). Thus, there was no consensus module relationship between males and females for the brown module, reflecting in the NA annotation in Figure 6c.

Gene ontology enrichment analysis showed that hundreds of protein-coding genes related to neuron projection, differentiation, nervous system development, and G-protein receptor signaling pathways were present in the brown module (Figure 7 and Appendix A). This confirms that, both in *S. mansoni* and *S. japonicum*, sexually mature males have a more active neuronal regulation than sexually mature females, including those mediated by G-protein signaling, which might point to a higher importance of neuronal processes in males during reproduction [21].

Furthermore, sperm motility and axoneme assembly, important for reproduction in males, were enriched in the brown module. Notably, these pathways were also found as enriched in one co-expression module in *S. mansoni* that has lncRNAs as hub genes and is more expressed in the testes [12]. The brown module has 1607 lncRNAs and 74 out of the 934 lincRNAs that our synteny analysis identified as possibly having orthologous lncRNAs in *S. mansoni*.

The brown module shares dozens of GO terms that were found to be enriched by Wang et al. [8] when they performed the GO analysis with the top 901 transcripts in *S. japonicum* males, with the highest positive correlation with the percentage of paired worms. Among the shared GO terms, it included synaptic vesicle endocytosis, cardiac muscle contraction, reproduction, and the amine transports activity that was further explored by Wang et al. [8]. It should be noted that in the original work [8], the authors did not identify that these transcripts had an opposite pattern of change as a function of sexual maturation in females (Figure 6d).

### 2.5. Expression of Genes Associated with Lipid Metabolism and Host Survival Increases Simultaneously in Males and Females

Additionally, we identified module eigengenes with the same pattern of expression, both in males and females as the post-infection days increased (Figure 6e—f); of note, the presence of patterns of simultaneous change in expression in both sexes was not explored by Wang et al. [8], who performed GO analyses with sets of genes that exclusively changed either in males or in females. The blue module, whose pattern of expression was positively correlated with the post-infection days in both sexes (Figure 6e), had a number of enriched GO terms associated with lipid and carbohydrates metabolism (Figure 8 and Appendix A). It is proposed that glycolysis provides energy sufficient for the survival of schistosomes, while vitellocytes are highly dependent on oxidative phosphorylation of lipids [22]. Schistosomes infecting mice living on high-fat diets have a higher fecundity rate than worms infecting mice living with a regular diet [23]. Vitellogenesis was also enriched in the blue module.

Other important pathways associated with survival inside the host were enriched (Appendix A), including drug metabolic process, cell activation involved in immune response, neutrophil-mediated immunity, and a negative regulation of the growth of symbionts in the host. The blue module had a total of 1520 lncRNAs, including 158 lincRNAs, with possible orthologs identified in *S. mansoni*.

### 2.6. The Expression of a Large Number of Genes Related to Neurogenesis is Repressed Simultaneously in Males and Females

Importantly, the turquoise eigengene represents genes whose pattern of expression decreased simultaneously in both sexes in the post-infection days (Figure 6f). The turquoise module eigengene showed a negative correlation coefficient with the post-infection days, both in females (−0.91) and in males (−0.96) (Figure 6a,b), and thus, displayed a negative consensus module relationship (−0.91) (Figure 6c). Most interestingly, GO analysis indicated that the turquoise module was enriched with genes related to the nervous system development and neurogenesis (Figure 9 and Appendix A).

The number of protein-coding genes associated with nervous system development in the turquoise module (810 genes) (Figure 9 and Appendix A) was much higher than the one present in the brown module (425 genes) (Figure 7 and Appendix A). This indicates that in *S. japonicum*, the genes that belonged to the brown module were related to nervous system development processes specific to males, being associated with axon projection and male–female signaling. Whereas, the genes belonging to the turquoise module were associated with the nervous system development that occurs in both sexes and must take place in the first weeks of infection. The latter processes seem to be mostly complete when all worms are paired and mature, as the genes related to these processes were massively turned off.

In this respect, it is known that the parasites pass through considerable changes in different tissues after entering the host, including in the nervous system [24], with some of these changes driven by stem cells in the schistosomula stage (first two weeks of infection) [25,26]. In *S. mansoni*, it was recently demonstrated that protein-coding genes associated with embryogenesis, neuronal development, brain development, and cell-fate differentiation, such as SOX, procadherin family, Wnt and frizzled receptors, have the highest expression in the sixth day post-infection, followed by a steady decline towards the adult stage [27]. In *S. japonicum*, the ortholog protein-coding genes were all present in the turquoise module: SOX, EWB00_011227 (mRNA16657); procharedin family, EWB00_002205 (mRNA3141), EWB00_00339 (mRNA4912), EWB00_009292 (mRNA13839); Wnt, EWB00_009528 (mRNA14229); frizzled receptor, EWB00_011205 (mRNA16607 and mRNA16608). Additionally, the turquoise module contained the neuroendocrine protein 7B2, EWB00_00462 (mRNA6788 and mRNA6789), which was identified as a marker gene for the cells of the cephalic ganglia in schistosomula obtained two days post-infection, in single-cell RNA-Seq experiments [26].

The turquoise was the largest module, with 8161 transcripts; 40% of which were lncRNAs, including 177 lincRNAs with possible orthologs identified in *S. mansoni*. Using the guilt-by-association approach we proposed that some of these lncRNAs might regulate the nervous system development processes.

### 2.7. Different Sets of Genes Related to Cell Replication and General Nucleotide Metabolism Have Their Expression Changed Only in One Sex Throughout the Sexual Maturation Process

We identified eigengene modules with correlation with the post-infection days only in one sex, such as the yellow and green modules, which had a positive correlation only in females (Figure 6a–c). The green module was more associated with general nucleotide metabolism (Appendix A), while the yellow module was mainly composed of genes associated with gene expression regulation and cell replication (Appendix A) pathways that were also enriched in *S. mansoni* gonads [12]. This pattern of expression was explored by Wang et al. [8], when they performed GO enrichment analysis with 645 transcripts in females, with the highest correlation for the percentage of females that developed vitellaria. Some of the GO terms identified by them were also present in the green and yellow modules enrichment.

The black module had no correlation with the post-infection days in females, had a negative correlation in males, and was enriched mainly with genes associated with meiosis, cell cycle regulation, DNA replication, and repair (Appendix A). The number of protein-coding genes in the pink and green-yellow modules were too small to produce reliable GO analysis. The modules in the red, purple, and magenta group had no GO enrichment that passed the statistical threshold. The grey group, comprised of genes that do not belong to any module, also had no GO enrichment.

Overall, the identification of lncRNAs co-expressed with the protein-coding genes within all modules points to a dynamic expression of lncRNAs, throughout sexual maturation in *S. japonicum*; these transcripts are correlated with the protein-coding genes related to schistosome survival and reproduction, which possibly regulate these processes.

## 3. Discussion

In this paper we reported the identification of 12,000 lncRNAs expressed in different life-cycle stages of *S. japonicum*, including cercariae, sporocysts, schistosomula, early-development, and adult males and females. Sequence similarity search and synteny conservation indicated that the lncRNAs of *S. japonicum* have synteny conservation with *S. mansoni* lncRNAs, even when there is a lack of sequence conservation. Pegueroles et al. [28] analyses of lncRNAs sequence or synteny conservation between the *Caenorhabditis* spp. also identified a much higher number of lncRNAs with synteny conservation than with sequence conservation [28].

We have shown that the lncRNAs in *S. japonicum* have a dynamic expression throughout sexual maturation in both males and females, being correlated to the genes involved with nervous system development, lipid and carbohydrate metabolism, drug metabolism, and parasite survival. This correlation can suggest that: (1) lncRNAs are regulated by the same activation program of the protein-coding genes; or (2) they regulate the expression of these protein-coding genes throughout the development in males and females. The role of lncRNAs regulating gene expression in sexual and tissue maturation in other species has been already reported in the literature [29,30,31]. Interestingly, it has been shown in amniotes that lncRNAs with dynamic expression patterns across developmental stages show signatures of enrichment for functionality, including a higher number of transcription factor binding sites in their promoters, in comparison with non-dynamic lncRNAs, thus, suggesting a stronger and more complex transcriptional regulation [32].

Finally, this study is the initial step towards the functional characterization of the role of *S. japonicum* lncRNAs in sexual maturation. Through synteny and co-expression analyses, good candidates can be prioritized for future functional studies. Validation by RT-qPCR of the expression of candidate lncRNAs of interest, at different developmental stages of the parasite, should be performed as the first step of functional characterization of these lncRNAs. Once the roles of lncRNAs are identified and given the clear importance of sexual maturation for both parasite reproduction and host immunopathogenesis, lncRNAs involved in these processes could be used as therapeutic targets. The lack of sequence similarity between *S. japonicum* and human lncRNAs, with only 19 lncRNAs having sequence similarity to human lncRNAs, is a very interesting feature for making them good candidates for therapies against schistosomes. As we have previously suggested, targeting the parasite lncRNA transcripts would have a reduced chance of unwanted off-target effects against the host [12]. Although the progression toward the clinic has been slow, lncRNAs represent potentially good therapeutic targets for human diseases [33,34,35].

## 4. Materials and Methods

### 4.1. Transcriptome Assembly and lncRNAs Classification

For this work, we used 66 *S. japonicum* RNA-Seq libraries that are publicly available at SRA from the following stages—cercariae, sporocysts, schistosomula, early-development, and adult males and females (Appendix A). The most recent SKCS01000000 version of the *S. japonicum* genome and transcriptome [13] were used as references.

The pipeline that was developed and applied by Maciel et al. [12] to identify lncRNAs in *S. mansoni*, was now applied to the above indicated *S. japonicum* RNA-Seq datasets; all parameters used in the pipeline outlined below were previously described in detail [12]. In brief, adapters and low-quality reads were removed by fastp [36] v 0.19.4, then, the reads were mapped against the genome with STAR [37] v 2.6.1c in a two-pass mode. RNA-Seq library strandedness was inferred by RSeQC [38] v 2.6.5 and this information was used in transcript reconstruction and expression level quantification. Multi-mapped reads were removed with Samtools [39] v 1.3 and uniquely mapped reads were used for transcript reconstruction for each library, with Scallop [40] v 10.2. A consensus transcriptome from all libraries was built using TACO [41] v 0.7.3.

Transcripts shorter than 200 nt, monoexonic, or with exon–exon overlap with protein-coding genes from the same genomic strand were removed from the set. The coding potential of the remaining transcripts was evaluated by means of the FEELnc tool [42] v 0.1.1, with a shuffle mode and by CPC2 tool [43] v 0.1. Only transcripts classified as lncRNAs by both tools were kept. The open reading frames (ORFs) were extracted from each putative lncRNA using ORFfinder v 0.4.3 (https://www.ncbi.nlm.nih.gov/orffinder/) and was submitted to annotation by the eggNOG-mapper webtool [44]. Transcripts with no hits against the eukaryote eggNOG database were then considered as lncRNAs.

When any transcript isoform was classified as a protein-coding mRNA at any step, all transcripts mapping to the same genomic locus were removed to avoid eventual pre-mRNAs. After this final step, a new GTF file was created containing the lncRNAs identified here, plus the protein-coding genes previously annotated by Luo et al. [13].

### 4.2. LncRNAs Conservation Analysis

To evaluate sequence conservation of lncRNAs from the *S. japonicum* across species, a search with the blastn tool [45] v 2.6.0 with a relaxed e-value cutoff of 1e-3 was performed against the sets of *S. mansoni* lncRNAs [12] and human lncRNAs [46] (GENCODE v 31).

Synteny analysis was performed to identify lncRNAs, which were contained inside syntenic blocks. In order to do that, genome, proteome, and CDS sequences in the FASTA format and the GFF3 protein-coding genes annotation from *S. japonicum* and *S. mansoni* were provided to the Synima pipeline [47]. The pipeline was run with default parameters, with OrthoMCL [48] v 1.4 as the method to identify orthologous protein-coding genes and DAGchainer [49], to identify the syntenic blocks. Bedtools [50] v 2.27.1 was used to compare the coordinates from the syntenic blocks identified by the Synima pipeline with the coordinates from the lncRNAs of *S. japonicum*.

Bedtools was also used to identify the closest protein-coding genes, upstream and downstream of the intergenic lncRNAs from *S. japonicum* and *S. mansoni*.

### 4.3. Weighted Gene Co-Expression Network Analysis

The new GTF file was used as the reference along with the genome sequence for mapping the reads of each RNA-seq library from Wang et al. [8], by using the STAR [37] v 2.6.1c, now in the one-pass mode, followed by gene expression quantification with RSEM [51] v 1.3. To reduce noise, transcripts with low expression (sum of counts < 10) were removed and the counts were transformed by Variance stabilizing transformation, using the vst function from the DESeq2 package [52] v 1.24.0. From the vst counts, we performed a multidimensional scaling (MDS) using the cmdscale R function using the Euclidian distance.

Differential eigengene network analysis was performed, using the Wang et al. [8] RNA-Seq dataset and the weighted gene co-expression network analysis (WGCNA) package [19] v 1.68. For consensus network construction, female and male counts were provided as two different sets in the blockwiseConsensusModules function (parameters power = 10, minModuleSize = 100, deepSplit = 2, mergeCutHeight = 0.25, networkType = signed).

Module preservation was plotted using the WGCNA package plotEigengeneNetworks function, with the days post-infection as the external information, as described in section II.5 from the WGCNA tutorials. webpage (https://horvath.genetics.ucla.edu/html/CoexpressionNetwork/Rpackages/WGCNA/Tutorials/). The equations behind the function are detailed by Langfelder and Horvath [20].

### 4.4. Gene Ontology (GO) Enrichment

Protein-coding genes identified by Luo et al. [13] were submitted to a GO annotation by an eggNOG-mapper [44]. Based on this annotation (Appendix A), we performed a GO enrichment analyses with BINGO [53] for the consensus modules identified by WGCNA. We used a hypergeometric test, the whole annotation as the reference set, and FDR ≤ 0.05 was used as the significance threshold.

## Figures and Tables

**Figure 1 ncrna-06-00015-f001:**
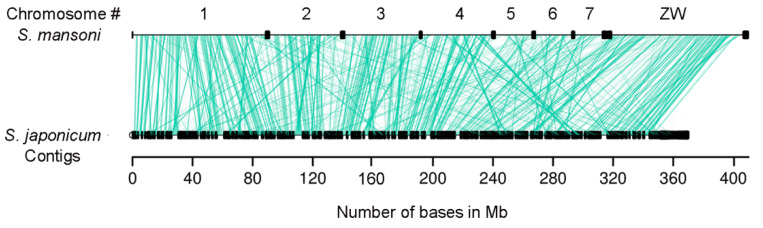
Synteny of protein-coding genes between *S. mansoni* and *S. japonicum* genomes. Syntenic blocks are indicated by the green lines connecting the genomes. Species names are shown on the left and the genomes are represented by horizontal black lines, with vertical black lines indicating scaffold/contig borders. For *S. mansoni* the chromosome number annotations of the longest scaffolds are indicated above the horizontal black line.

**Figure 2 ncrna-06-00015-f002:**
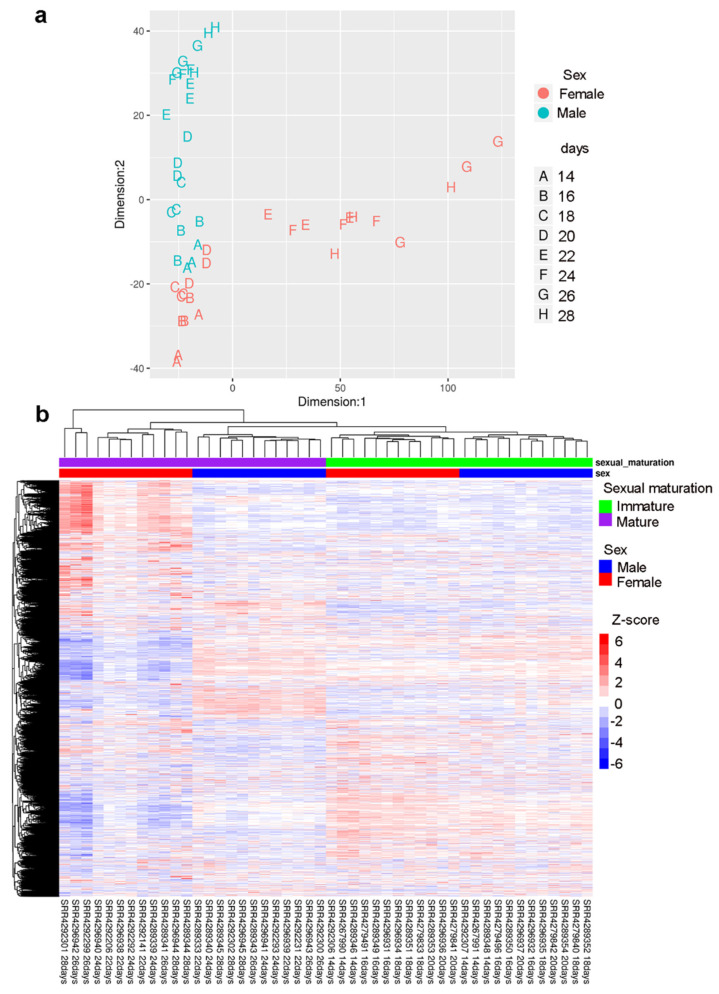
Protein-coding and lncRNA expression profile of *S. japonicum* throughout sexual maturation. (**a**) Multidimensional scaling (MDS) analyses of protein-coding and lncRNA gene expression together, detected by RNA-Seq in 48 samples of *S. japonicum* males (**blue**) and females (**red**) perfused from mice at 14 to 28 days post-infection. The number of days post-infection are indicated by the letters A to H, according to the legend on the right; three replicate samples for each day were analyzed. (**b**) Gene expression heatmap of protein-coding and lncRNA genes together (each in one line), across all 48 samples (each in one column) as indicated by the sample ID label at the bottom. Unsupervised clustering using the Euclidean distance was performed; expression of each gene is shown as the z-score (from −6 to 6), which is the number of standard deviations above (**red**) or below (**blue**) the mean expression value of that gene, across all RNA-Seq libraries; the z-score color scale is shown on the right. Samples from days 14 to 20 are labeled at the top as immature parasites (**green**), and from days 22 to 28 as mature parasites (**purple**). Males are labeled at the top in blue and females in red.

**Figure 3 ncrna-06-00015-f003:**
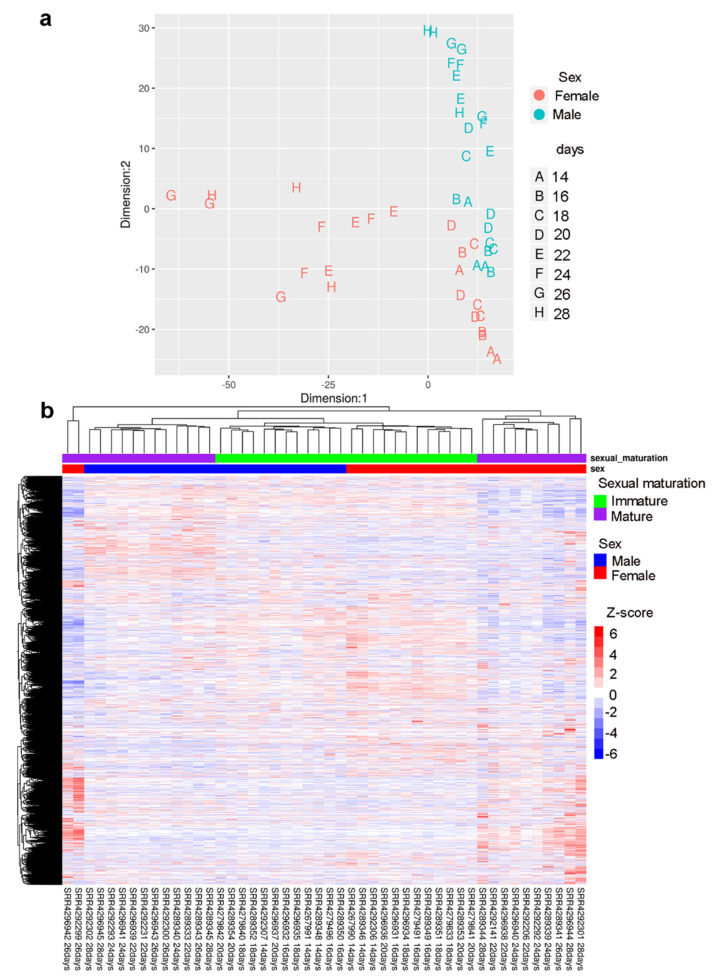
LncRNA expression profile of *S. japonicum* throughout sexual maturation. (**a**) Multidimensional scaling (MDS) analyses of lncRNA gene expression alone, detected by RNA-Seq in 48 samples of *S. japonicum* males (**blue**) and females (**red**) perfused from mice at 14 to 28 days post-infection. The number of days post-infection are indicated by the letters A to H, according to the legend at right; three replicate samples for each day were analyzed. (**b**) Gene expression heatmap of lncRNA genes (each in one line), across all 48 samples (each in one column) as indicated by the sample ID label at the bottom. Unsupervised clustering using the Euclidean distance was performed; expression of each lncRNA gene is shown as the z-score (from −6 to 6), which is the number of standard deviations above (**red**) or below (**blue**) the mean expression value of that gene across all RNA-Seq libraries; the z-score color scale is shown on the right. Samples from days 14 to 20 are labeled at the top as immature parasites (**green**), and from days 22 to 28 as mature parasites (**purple**). Males are labeled at the top in blue and females in red.

**Figure 4 ncrna-06-00015-f004:**
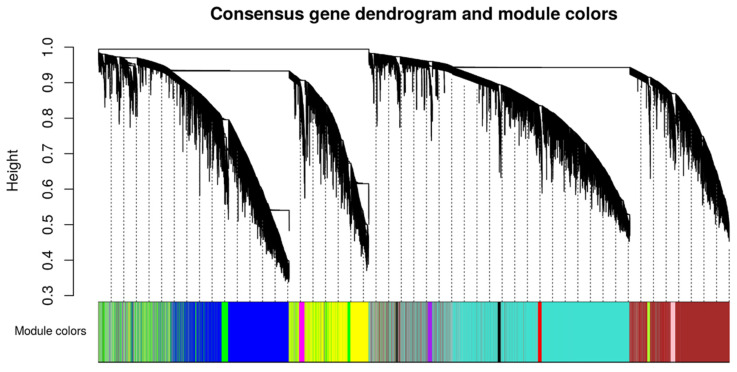
Consensus modules identified by weighted gene co-expression network analysis (WGCNA). Gene hierarchical cluster dendrogram based on a dissimilarity measure of the Topological Overlap Matrix (1–TOM) calculated by WGCNA; the color labels correspond to the different gene co-expression consensus modules identified between male and female datasets.

**Figure 5 ncrna-06-00015-f005:**
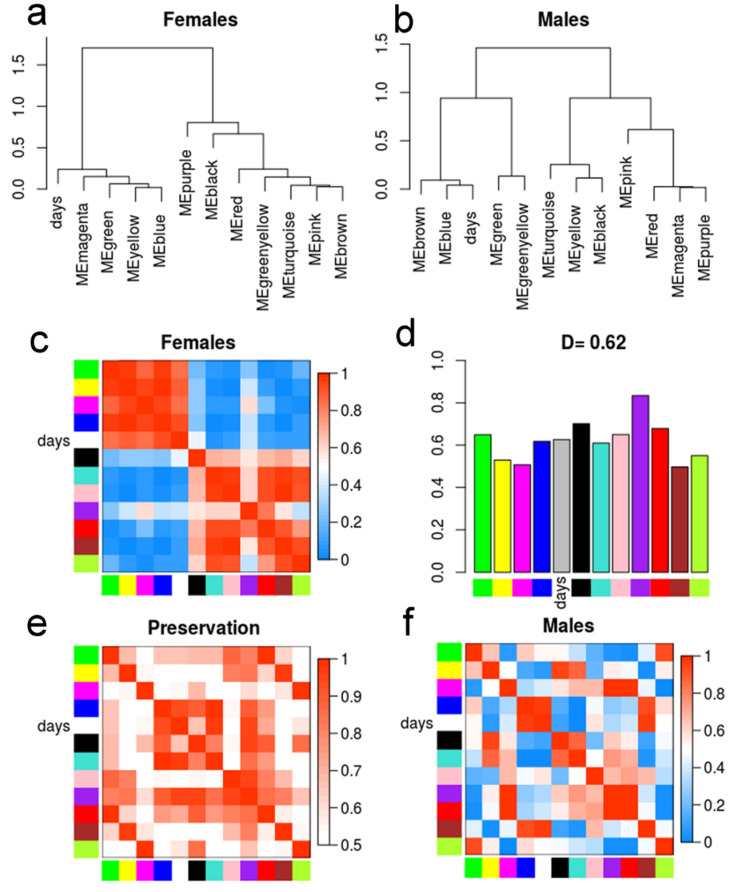
Differential eigengene network analysis. (**a**,**b**) Clustering dendrograms of the consensus module eigengenes for females and males, respectively. (**c**,**f**) Females (**c**) and males (**f**) heatmaps of eigengene adjacencies (correlation matrix) in the consensus module eigengenes network. Each row and column correspond to one of the eleven module eigengenes (colors). The correlation between the eigengene and the number of days post-infection (days, white) is also indicated. The cells are colored according to the scale at right; red indicates high adjacency (positive correlation) and blue indicates low adjacency (negative correlation). (**d**) Preservation measure for each consensus eigengene. Each colored bar corresponds to the eigengene of the corresponding module color. The *y*-axis gives the eigengene preservation measure. The D value denotes the overall preservation of the eigengene networks. (**e**) Heatmap of adjacencies in the preservation network between females and males. Each row and column correspond to a consensus module; saturation of the red color is proportional to preservation, according to the color legend.

**Figure 6 ncrna-06-00015-f006:**
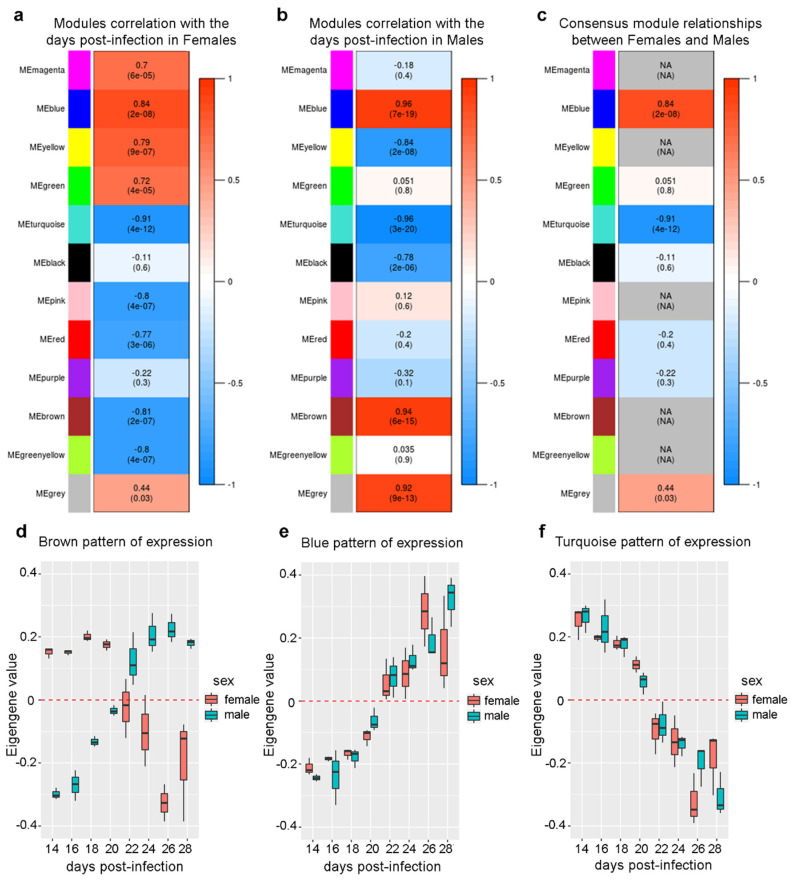
Relationship between the module eigengenes (ME) and the days post-infection. (**a,b**) Module eigengenes (ME) correlation with the days post-infection in females (**a**) and males (**b**). Each row in the tables corresponds to a module; the number shows the correlation of the corresponding module eigengene with the days, *p*-values given in parentheses. Tables are color-coded by correlation according to the color scales in the right. (**c**) Consensus module relationships between the female and male module eigengenes and the days post-infection. Consensus relationship correlation is shown when both go in the same direction; missing (NA) entries indicate that the correlations in the male and female datasets have opposite signs. (**d**–**f**) The boxplots indicate the value of the first principal component, called eigengene, for the samples of males (**blue**) and females (**red**) from 14 to 28 days post-infection for the modules brown (**d**), blue (**e**), and turquoise (**f**).

**Figure 7 ncrna-06-00015-f007:**
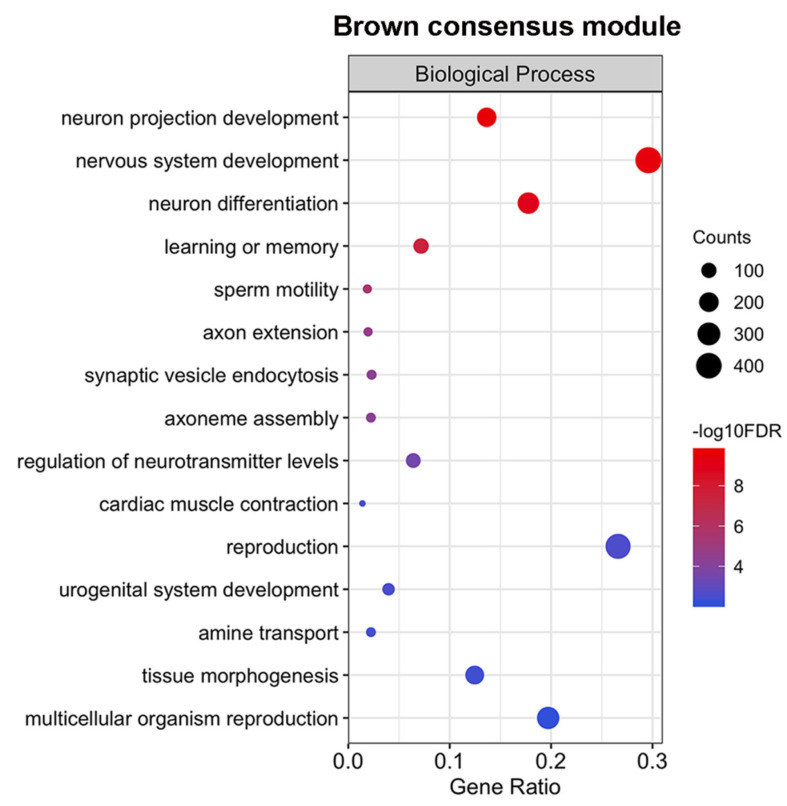
Selected fifteen Gene Ontology enriched terms for protein-coding genes of the brown consensus module. Only GO terms from the Biological Process category are represented. The size of the circles is proportional to the number of genes (**counts scale on the right**) in each significantly enriched GO term, and the colors show the statistical significance of the enrichment, as indicated by the –log10 FDR values (**color-coded scale at the right**).

**Figure 8 ncrna-06-00015-f008:**
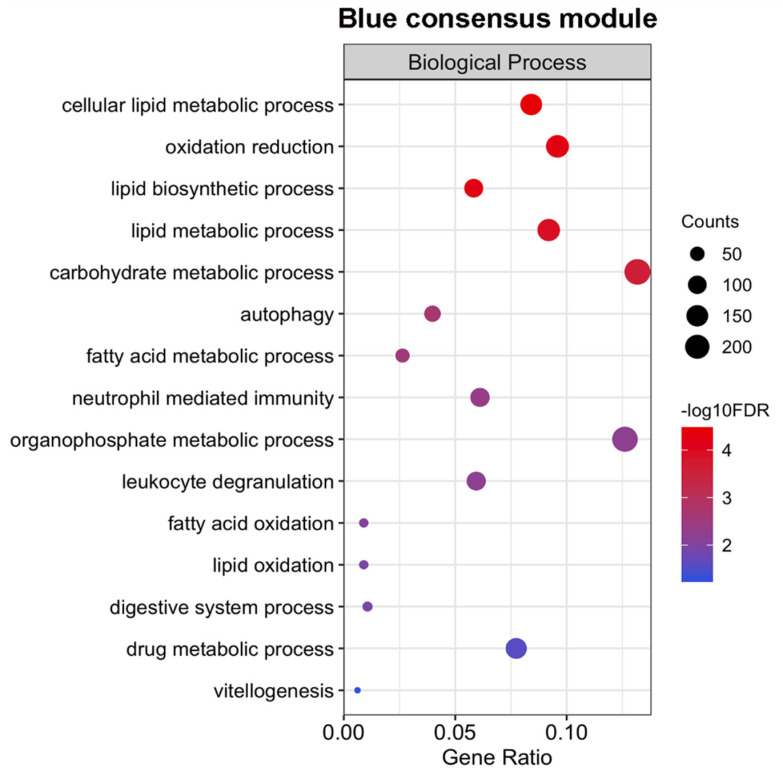
Selected fifteen Gene Ontology enriched terms for protein-coding the genes of the blue consensus module. Only GO terms from the Biological Process category are represented. The size of the circles is proportional to the number of genes (**counts scale on the right**) in each significantly enriched GO term, and the colors show the statistical significance of the enrichment, as indicated by the –log10 FDR values (**color-coded scale at the right**).

**Figure 9 ncrna-06-00015-f009:**
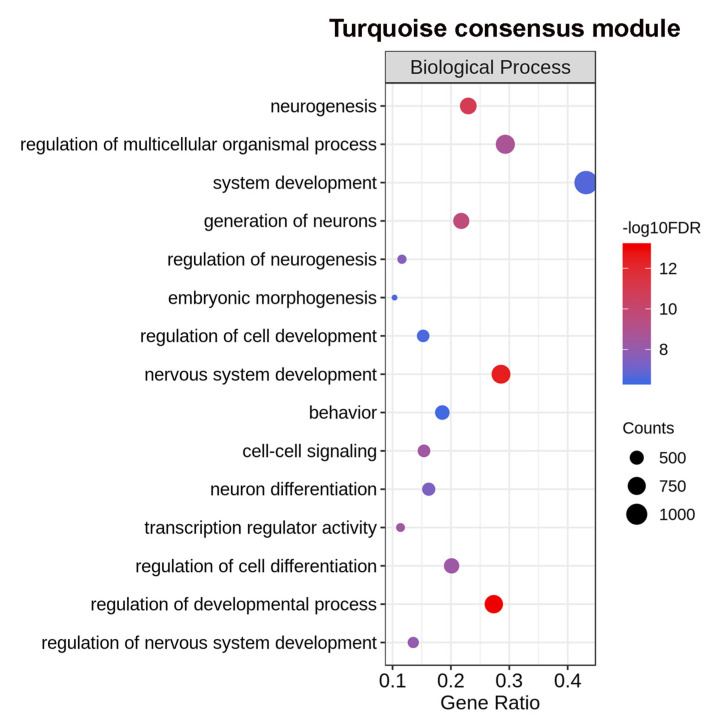
Top fifteen most significantly enriched Gene Ontology terms from the Biological Process category for protein-coding genes of the turquoise consensus module. On the left are the enriched GO term annotations. The size of the circles is proportional to the number of genes (**counts scale on the right**) in each significantly enriched GO term, and the colors show the statistical significance of the enrichment, as indicated by the –log10 FDR values (**color-coded scale on the right**).

**Table 1 ncrna-06-00015-t001:** Number of transcripts per module and percentage of lncRNAs in each module.

Module Color	Total Number of Transcripts	Number of lncRNAs	% of lncRNAs
Black	336	82	24
Blue	4465	1520	34
Brown	4400	1607	37
Green	2288	960	42
Green-yellow	128	53	41
Magenta	219	91	42
Pink	288	98	34
Purple	174	75	43
Red	521	260	50
Turquoise	8161	3300	40
Yellow	2830	877	31
Total	23,810	8923	37

## Data Availability

The data sets analyzed in this study can be found in the SRA repository (https://www.ncbi.nlm.nih.gov/sra). The specific accession numbers for each and all data sets that were downloaded from these databases and used here are given in Appendix A. The GTF containing the lncRNAs annotated in this work are available at http://schistosoma.usp.br/.

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
