# Peer review of "Dynamic Expression of Long Non-Coding RNAs Throughout Parasite Sexual and Neural Maturation in Schistosoma Japonicum"

_ncrna, 2020, doi:10.3390/ncrna6020015_

Round 1

Reviewer 1 Report

In this manuscript Maciel et al. present a detailed descriptive study of the expression of long non-coding RNAs throughout parasite sexual and neural maturation in Schistosoma japonicum. The re-analysis of available data allows the authors to provide a description of how the lncRNAs are expressed in a thorough manner, which would be of interest for the field and the readers of the journal. 

No major comments, however I believe that some validations by qPCR at different stages of the cycle may improve the manuscript and the conclusions. 

Author Response

We thank the reviewer for the positive comments and for the constructive suggestion. We have now added to the Discussion section of the revised version the following sentence:

Validation by RT-qPCR of the expression of candidate lncRNAs of interest, at different developmental stages of the parasite, should be performed as the first step of functional characterization of these lncRNAs. (please see lines 421-423)

Reviewer 2 Report

Authors presented an elegantly bioinformatic data analysis of mRNAs and lncRNAs Schistosoma japonicum, based on RNAseq libraries and revisiting the published data. The work enrich the analysis of lncRNA, clarifing the missidentified lncRNA and its conserved expression inside the orthologous syntenic blocks in S. mansoni.
Authors explore the pathway analysis using GO to show the top pathways enriched in the gene-sets of the modules representative for differential expression analysis by WGCNA, linking pathways and maturation stage in each gender.
I recommend the article publication.

Author Response

We thank the reviewer for the positive comments. As requested, minor spelling corrections have been performed.

Reviewer 3 Report

This manuscript by Maciel et al is thorogh evaluation of lncRNAs from S. japonicum. The analyses are in depth and provide interesting information to scientists interested in male-female interactions. The manuscript is well written and the experimental design follows a logical progression. e are some minor concerns below.

Line 61 schistosome delete s

Lines 147/294/379; Wang et al citation 8 does not list all the authors but the cited references list all the authors eg line 185?

Line 470; space between plot eigengene Network

Author Response

We thank the reviewer for the positive comments. All spelling errors and reference citation errors have been corrected as follows:

(1) Line 61 schistosome delete s  

As requested, schistosomes has been changed to schistosome (see line 62)

(2) Lines 147/294/379; Wang et al citation 8 does not list all the authors but the cited references list all the authors eg line 185? 

As requested, the citation format has been corrected to include all authors. Please see the corresponding new lines 150/297/385.

(3) Line 470; space between plot eigengene Network 

Please note that this is a WCGNA package function, which is written without spaces. We have now re-written the sentence to make it clearer. Please see line 490.